# Thio-NHS esters are non-innocent protein acylating reagents

Weibing Liu [1,2,5], Aziz Khan[1,3,5], Yana Demyanenko [1,5],
Shabaz Mohammed [1,3,4] ✉ & Benjamin G. Davis [1,2,3] ✉

*N*-Hydroxysuccinimide (NHS)-ester derivatives are widely used reagents in biological chemistry and chemical biology. Their efficacy relies critically on the exclusive chemoselectivity of activated acyl over that of the imidic acyl moieties in the succinimide. Here, through systematic structural variation that modulates acyl reactivity, coupled with a statistically controlled ultra-rapid screen for unknown modifications in tandem mass spectra as well as lysine profiling across complex lysine environments, including those within proteomes containing many thousands of proteins, we reveal that ring-opening to afford *N*-succinamide derivatives is a present, sometimes dominant, side-reaction. The extent of side-reaction is shown to be site-dependent, with side-reaction and desired reaction occurring within the same protein substrate. The resulting formation of bioconjugates with unintended, unstable linkages and modifications suggests the re-evaluation of: (i) known commercial reagents; and (ii) functional conclusions previously drawn using NHS esters in areas as diverse as antibody-drug biotherapy, vaccination and cross-link-enabled structural analyses.

*N*-Hydroxysuccinimide (NHS)-esters are ubiquitous reagents for acyl bond formation[1–5]. Of the diverse applications exploiting their broad (including commercial[6,7]) availability, they have long been suggested in protein labelling to modify the nucleophilic sidechains not only of lysine, but also cysteine, serine, threonine and tyrosine[2,8–10]. They have therefore seen long-standing use to produce functional modification of biomolecules by, in particular, exploiting *N*-terminal or Lys ε-amino reaction to form stable amide bonds via release of the NHS (or sulfo-NHS) group[11,12].

Thousands of commercially-available NHS esters and new reagents based on NHS chemistry therefore exist as versatile tools for the preparation of bioconjugates bearing, for example, labels, dyes, affinity motifs, reactive groups and therapeutic payloads[13,14] Notably, NHS-esters are seen as possessing the requisite properties, including rapid reactivity and excellent biocompatibility, to benignly enable the industrial generation of therapeutic biologics, such as immunogens[15–19] or drug-targeting conjugates[20]. Furthermore, their predictable

reactivity has led to their adoption as precise tools with faithful reactivity for probing the conformation and function of proteins through intra- and inter-molecular chemical cross-linking[21–24]. Both of these applications, amongst others, represent examples where the precise nature of the conjugate (and its linker e.g. immunogenicity in human use or inferred length in cross-link-enabled structural mapping) plays a critical role. To date, it has near universally been assumed that only direct acylation products dominate.

Here, through a systematic survey of a family of NHS esters and the testing of their reactivity across a diverse range of proteins, we now identify non-innocent bioconjugation side-products, including those resulting from the 'ring-opening' of the succinimide moiety within the NHS ester (Fig. 1a) that question the universality of these assumptions.

## Results

One primary industrial[25] and academic[7] use of NHS esters for over four decades is in the preparation of thiolated protein substrates using

[1]The Rosalind Franklin Institute, Harwell Oxfordshire, UK. [2]Department of Pharmacology, University of Oxford, Oxford, UK. [3]Department of Chemistry, University of Oxford, Oxford, UK. [4]Department of Biochemistry, University of Oxford, Oxford, UK. [5]These authors contributed equally: Weibing Liu, Aziz Khan, Yana Demyanenko. ✉e-mail: Shabaz.Mohammed@rfi.ac.uk; Ben.Davis@rfi.ac.uk

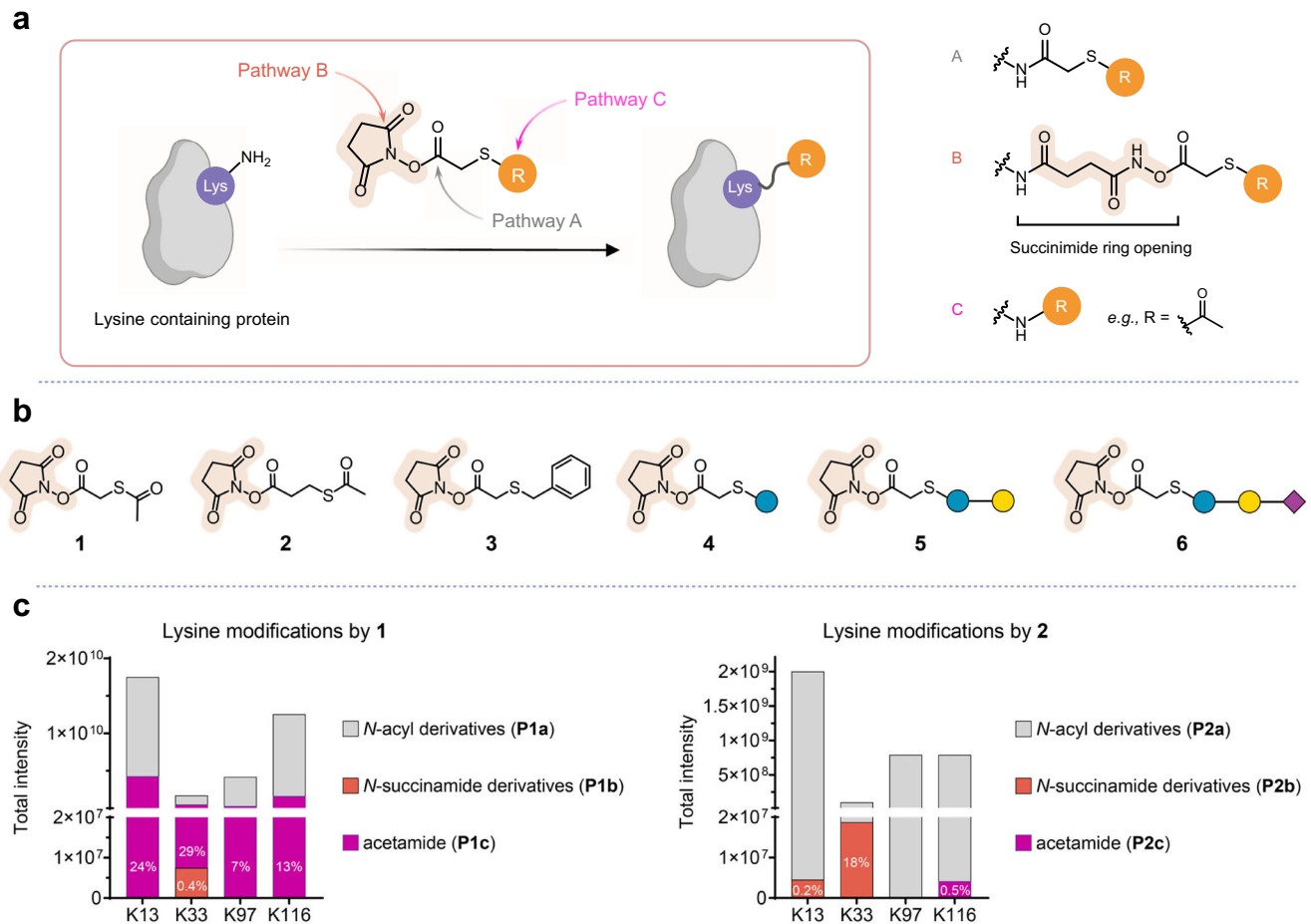

**Fig. 1 | Non-innocent behaviour of thio-NHS esters. a** The reactivity of ε-amino groups within lysine extends beyond canonical pathway A, involving nucleophilic substitution reactions with the oxocarbonyl moiety of succinimide to generate *N*-acyl derivatives. Pathway B, a non-innocent route, is characterised by the engagement of amine nucleophiles in nucleophilic substitution reactions with imidic acyl moieties in the succinimide, previously presumed simply to be an innocent 'leaving group'. This leads to the production of *N*-succinamide derivatives, via the mechanism of succinimide ring opening. Pathway C, entails thioester formation with R as an acyl group in the NHS ester, leading to acylated derivatives through nucleophilic substitution at the thioester's carbonyl group. Parts created in BioRender: Govender kirkpatrick, M. (2025) https://BioRender.com/8olwrd6.
**b** Structures and reactions of SATA **1**, SATP **2**, S-benzyl-NHS-ester **3**, Glc-NHS **4**, Gal-Glc-NHS **5** and Sia-Gal-Glc-NHS **6**. **c** Reaction pathway chemo- and regio- selectivities are site-dependent. The peptide abundances of different lysine modifications induced by SATA **1** and SATP **2** reveal site dependency that determines not only rate and extent of reaction but even pathway. Products of pathway A are labelled in grey, while alternative pathways B and C are highlighted in orange and magenta, respectively.

commercially-available[26] reagents such as so-called SATA **1** or SATP **2** or for the attachment of payloads (such as cytotoxins[27]) derived from the same NHS-activated-thioalkanoyl substructure. Reagents such as **1** and **2** are somewhat unusual in that they bear two potentially-activated acyl groups: as well as the activated acyl moiety in the NHS ester moiety of these reagents, they also contain a thioester that also has the potential[28–30] to act as an acyl donor. As part of a project to evaluate the relative chemoselectivity of these two acylating moieties (Fig. 1a, 'wanted' path A *versus* unwanted path C, respectively) we tested their protein reactivity in depth. The increasing combination of high performance, rapid, mass spectrometers and 'cheap' computational power with the advent of highly efficient search engines that can perform semi-agnostic, error-tolerant, localisation-aware analyses[31] now raises the potential to more fully identify protein-chemistry reaction products and so localise unexpected side-products from MSMS data without prior knowledge of the exact mass or molecular composition of the modification. This now creates the potential to enable effective pipelines for swift characterisation of protein chemistry; NHS-ester protein chemistry is an archetypal benchmark.

Consistent with the potential for dual modes of acylation, archetypal protein model hen-egg lysozyme (HEL, containing six lysine (Lys)

residues) was reacted with **1** and **2** and examined by proteolysis and LC-MS/MS analysis. To comprehensively investigate the modification of lysine residues, we utilised a modification and residue agnostic search approach[31,32]. The results revealed mixed pathways, indicating both the NHS-ester-mediated acylation path A (wanted) as well as direct lysine acetylation as a side-product originating from path C. However, critically enabled by semi-agnostic analyses, we also surprisingly observed the formation of *N*-succin*amide* derivatives **P1b, P2b**. These adducts, seen even at levels surpassing that of side pathway C (Fig. 1c, Supplementary Fig. 1), arose from a wholly unexpected pathway B that involved the presumed chemically innocent succinimide leaving group of NHS-esters instead undergoing ring opening (Supplementary Figs. 1, 2).

Since HEL contains six lysine residues, in these intact-protein mass spectra multiple peaks corresponding to the coupling of one to six lysine residues were identified, each of which lie in different environments in the protein. In principle, therefore, each residue might react via each pathway with different rates driving multiple regio- and chemo-selective reaction outcomes. To probe this, we used **1** and **2** that can react potentially via any of the three pathways A, B or C, by surveying, using semi-agnostic MSMS analyses, all of the 216 possible permuted products (Fig. 1c, Supplementary Figs. 1, 2). For both **1** and **2**, reactions at

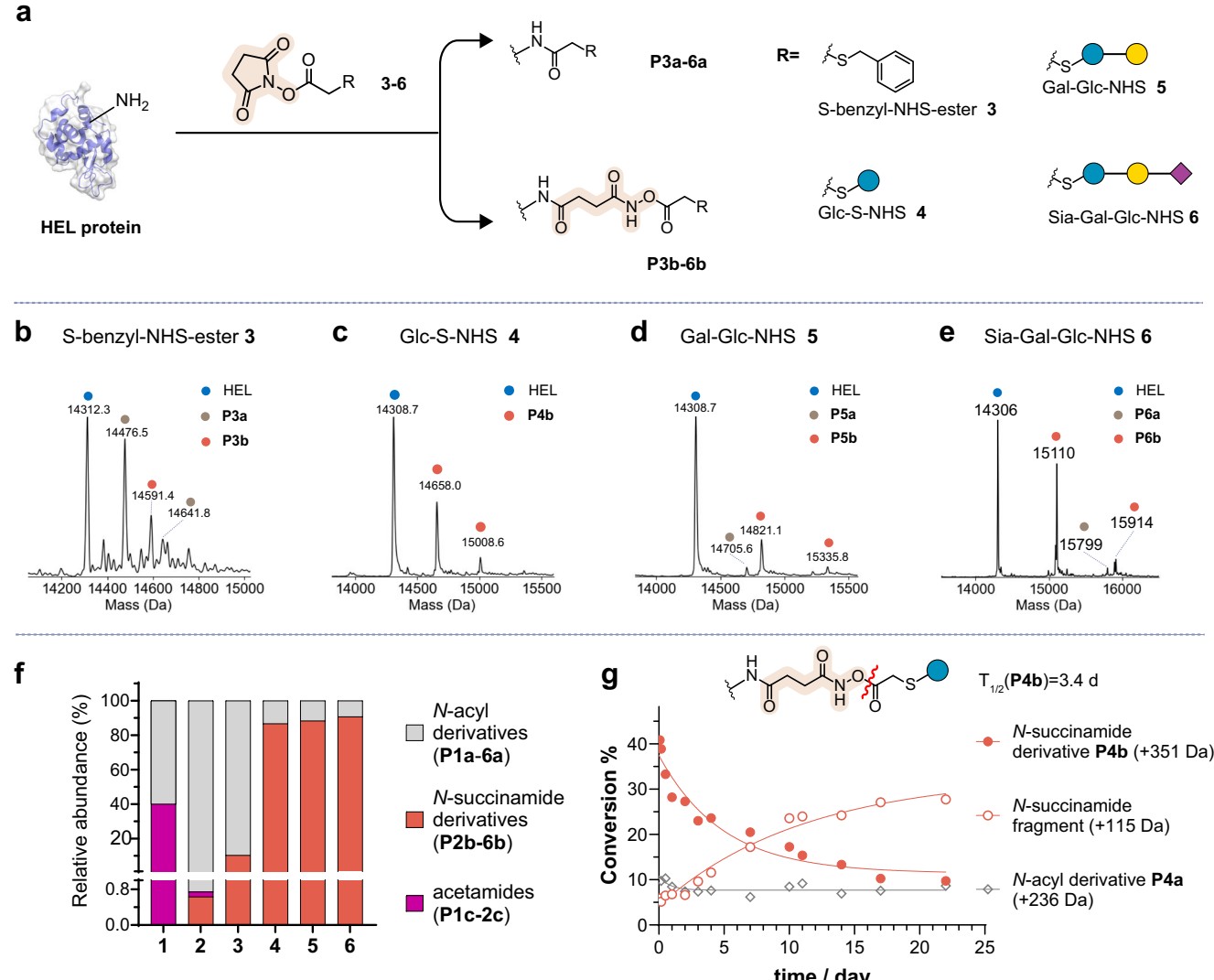

**Fig. 2 | Evaluation of the reaction of HEL protein with systematically varied thio-NHS esters. a** NHS esters with increasing structural complexity of α-thiosubstitution S-benzyl-NHS-ester **3**, Glc-NHS **4** and Gal-Glc-NHS **5** give rise to *N*-acyl derivatives (**P3a-6a**) and *N*-succinamide derivatives (**P3b-6b**). **b–e** LC-MS analysis of the covalent modification of the HEL protein by the three thio-NHS esters. Starting material is labelled with blue, while pathway A and B products are labelled in grey and orange, respectively. Reagents and conditions: 200 equivalents of **3**, **4**, **5** or **6** (respect to Lys), 1 mg/mL HEL, 0.1 M NaHCO₃ in PBS, 0 °C, 2 h.

**f** Prevalence of products of each of the three reaction pathways for compounds **1–6** (A−Grey, B−Orange, C−Magenta). Comprehensive analysis of all possible variants of each modification was performed; the intensities of all peptides modified via each pathway were summed to yield the final ratios (see Supplementary Table 1 and Experimental Procedures). **g** Stability of the *N*-acyl derivatives (**P4a**) and *N*-succinamide derivatives (**P4b**). **P4b** decreases over time due to ester bond cleavage, generating fragments with a mass increase of +115 Da.

four lysine residues (K13, K33, K97 and K116) were observed. K97 displayed most consistent canonical pathway A chemoselectivity for both reagents (>93% pathway A), whereas K33 displayed the highest level of noncanonical reactivity (71–82% pathway A) strikingly indicating that single NHS reagents display not only inherently different *chemo*-selectivity with respect to Lys residues in a given protein (different reactive pathways) but also marked *regio*-selectivity within those pathways for certain Lys residues. At different sites, markedly different *chemo*-selectivity for different pathways A, B and C was observed. Strikingly with **2**, non-canonical pathway B *N*-succinamide derivatives were observed exclusively on K13 and K33, with K33 showing a significant proportion (18%) of *N*-succinamide modification, whereas non-canonical pathway C acetamide formation was exclusively detected on the K116 residue. Together, these data highlight the extent to which site alone can determine dramatic variation in not only the level of reactivity (driving regioselectivity) but even the type of reaction (driving chemoselectivity) down unwanted paths. It is notable that although global levels of

*N*-succinamide with **2** in HEL were low (0.6%), at a single site this ring-opening pathway rose relatively to 18%.

To systematically probe the structural range of NHS esters that might also exhibit ring-opening pathway B reactivity (Fig. 1) we synthesised (Fig. 1b, Supplementary Figs. 3, 4, 5) four distinct types of NHS-esters of increasing structural complexity from the substructures found in **1**: S-benzyl-NHS-ester **3** and three systematically extended mono-, di- and then tri-saccharide NHS-esters that allow synthetic glycoprotein formation, Glc-NHS **4**, Gal-Glc-NHS **5** and Sia-Gal-Glc-NHS **6**. Under essentially identical conditions, LC-MS revealed increasing levels of ring-opening pathway B with increasing structural complexity (Fig. 2, **P6b**, **P5b**, **P4b** > **P1b**). Indeed, even via intact-protein MS1 analyses, despite the complexity of statistical modification mixtures, distinct peaks corresponding to *N*-succinamide derivatives could be observed (Fig. 2b–e and Supplementary Figs. 7 and 8). Notably in the more complex glycoprotein samples **P6b**, **P5b**, **P4b** pathway B dominated over canonical *N*-acyl pathway A, in some cases even up to ~90% of total content (Fig. 2f).

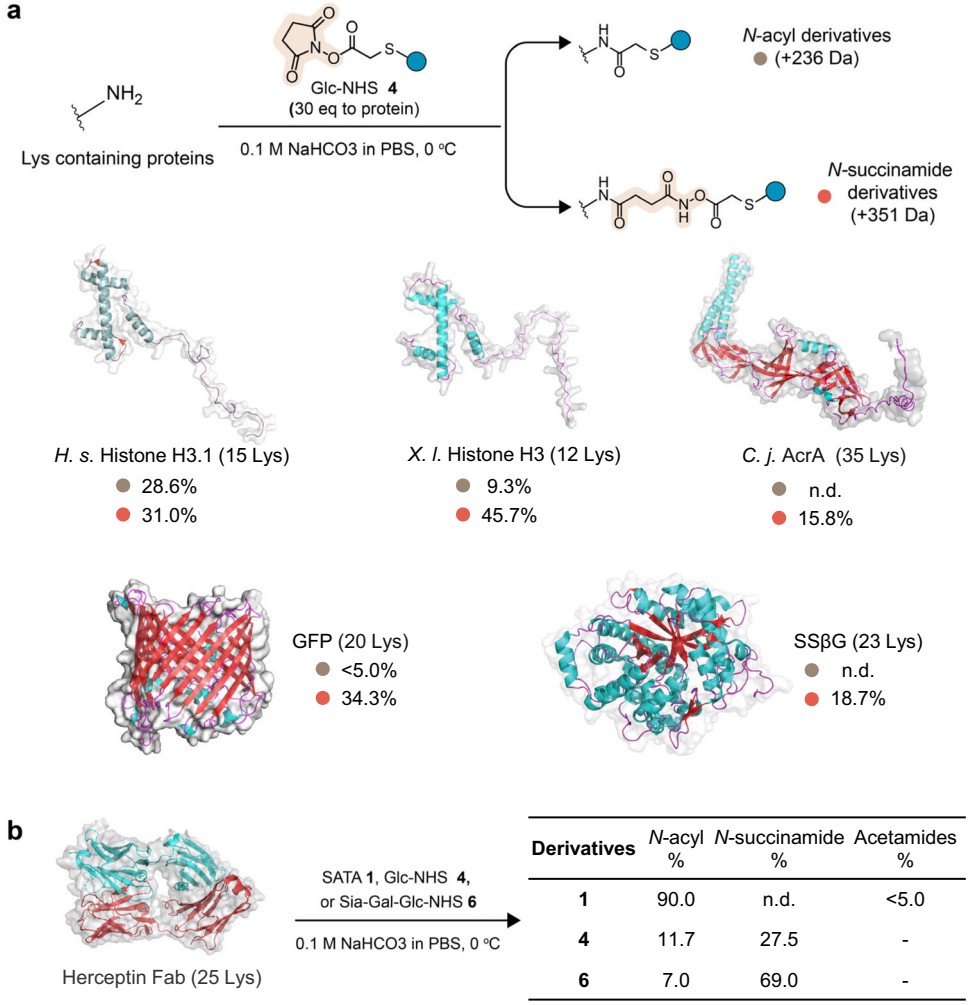

**Fig. 3 | Evaluation of the reaction of thio-NHS esters with different protein substrates. a** Diverse proteins with different secondary structural motifs illustrated via surface and cartoon models showing the distribution of alpha-helices (cyan) and beta-sheets (red) and the different lysine environments that give rise to *N*-acyl derivatives (grey) and *N*-succinamide derivatives (orange). **b** LC-MS analysis of reaction between Herceptin and SATA **1**, Glc-NHS **4**, Sia-Gal-Glc-NHS **6**. n.d. Not detected by intact-protein MS1 analysis. The structure shown is the Fab region of Herceptin, with yield calculated based on the light chain (shown in red). Conditions: 200 equivalents of **1**, **4** or **6** (respect to proteins), 0.5 h for **1** and **4**, 2 h for **6**.

Given the potential for varied use of NHS esters as commercially available reagents, we also tested whether outcome of the reaction was significantly influenced by varying reaction conditions. Notably, regardless of the equivalents used or the reaction solvent, **6** consistently afforded the major ring-opened product of *N*-succinamide derivative **P6b** (Supplementary Fig. 9). Higher equivalents of **6** simply increased the yield of the two products **P6a** and **P6b**, while higher concentration of HEL was detrimental to the conversion. Highest conversions were observed when using 0.1 M bicarbonate solution, while no significant difference in product yield was observed when using DMF or DMSO as commonly used co-solvents (Supplementary Fig. 9). We also evaluated the impact of different temperatures on the ring-opening reaction pathway by performing the reaction at both 37 °C and on ice; there was minimal difference (Supplementary Fig. 9). Together these results suggest that when pathway B ring-opening of succinimide is present it is likely independent of common variations of reaction conditions. Notably, after initial formation, pathway B ring-opening products (**P4-6b**) decrease with prolonged incubation (Supplementary Figs. 10 and 11), with most of **P4b** converting to a +115 Da fragment and exhibiting a half-life of ~3.4 days (Fig. 2g), indicating an instability of the extended succinamide conjugate that appears to decrease with increasing structural complexity.

We gauged reactivity in proteins containing more lysines (from 12 to 35), using a variety of representative protein folds and functions: the small α-helical nuclear histone H3 proteins from *H. sapiens* and *X. laevis*, the β-strand *C. jejuni* glycoprotein AcrA, the β-barrel green fluorescent protein GFP, and (αβ)₈-barrel glycosidase enzyme SSβG. All showed ring-opening reactivity giving rise to pathway B *N*-succinamide derivative modification (Fig. 3a and Supplementary Fig. 12). Reactions (Fig. 3b and Supplementary Fig. 13) of **1**, **4** and **6** with Herceptin (as a model therapeutic IgG antibody that continues to be used as pivotal platform to generate antibody-cytotoxin conjugates[33,34] with varying success[35]) also mirrored our findings with other proteins: with **4** and **6** the unwanted ring-opening pathway B adduct was the major product. These results suggest that the occurrence of non-canonical pathways B and C (ring-opening or acetylation) are a more general phenomenon and not specific to a particular protein substrate.

To more comprehensively understand whether the lysine microenvironment plays a critical role in driving such ring-opening reactions, we then profiled reactive lysines within highly diverse complex protein samples. Thus, the proteomes of disrupted prokaryotic cells (released from their native contexts) were used to assess the environmental and dynamic reactivity of **1**, **2** and **4**. *E. coli* (K12 strain) cells were lysed under physiological conditions and treated with **1**, **2** and **4**, along with a control group treated with vehicle, followed by analysis

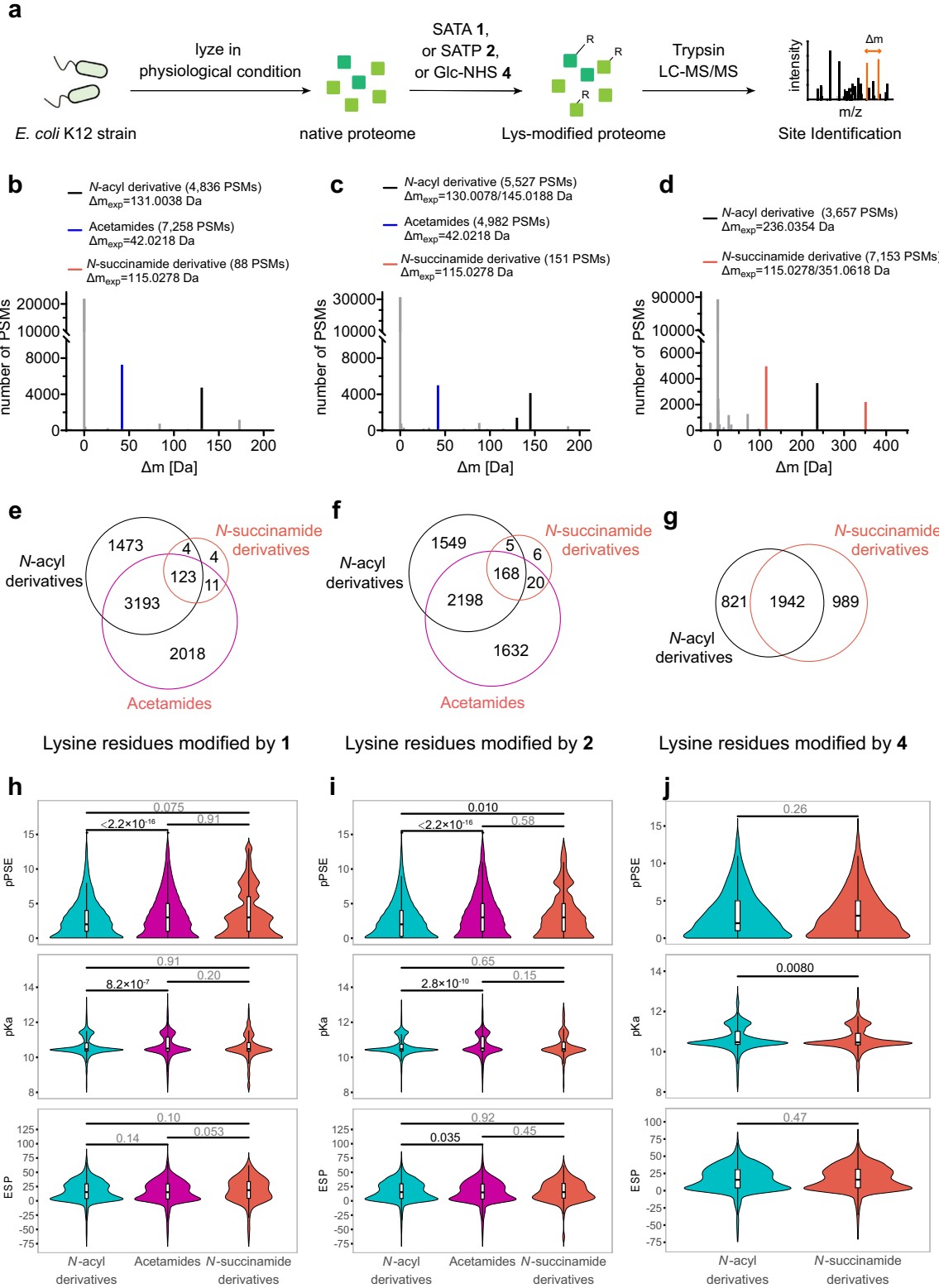

for site modifications (Fig. 4a). Using an open search method[31,32], we identified the distribution of mass deviations arising from ring-opening pathway B modifications from **1**, **2** and **4**, as well as pathway *N*-acyl modifications from **1**, **2** and **4**, and acetamide pathway C modifications from **1** and **2**; these represented the major modifications occurring proteome-wide (Fig. 4b–d). We then conducted a closed

search to explore masses with the $\Delta m_{exp}$ as the offset mass (see *Methods*). Following thorough data filtering, significant levels of ring-opening pathway B *N*-succinamide formation was observed (Fig. 4e–f and Supplementary Fig. S14a) in a background of pathways A and C. Notably, we also identified that a substantial number of lysine residues reacted through two or even three of these pathways (Fig. 4e–g).

**Fig. 4 | Proteomic analysis of the reaction of thio-NHS esters with entire native prokaryotic proteome. a** Proteomic mapping of both proteins and modification sites for reaction with SATA **1**, SATP **2** and Glc-NHS **4**. **b–d** Masses of modification that occur proteome-wide by **1**, **2** and **4**, respectively. The black peaks indicate *N*-acyl modifications, the orange peaks indicate *N*-succinamide modifications, and the blue peaks indicate acetamide modifications. PSMs peptide spectrum matches. (**e–g**) Venn diagram showing the overlap between lysines of different modifications by **1**, **2** and **4**, respectively. **h–j** Violin plot comparing the distribution of lysines of different modifications by **1**, **2** and **4**, respectively for accessibility, charge and polarity. AlphaFold-enabled pPSE (prediction-aware part-sphere exposure)[36], pKa and ESP (electrostatic potential) are predicted measures of accessibility, charge and polarity. Sample sizes (lysine sites per group): **h**–*N*-acyl ($n = 4491$, cyan), acetamides ($n = 5094$, magenta), *N*-succinamide ($n = 138$, orange); **i**–*N*-acyl ($n = 3648$), acetamides ($n = 3851$), *N*-succinamide ($n = 191$); **j**–*N*-acyl ($n = 2603$), *N*-succinamide ($n = 2758$). Low-confidence predictions (prediction quality ≤ 70) were removed. Statistical significance was determined using the two-sided Wilcoxon rank sum test with Benjamini–Hochberg adjustment for multiple comparisons. Exact *p* values are shown.

To gain further mechanistic insight, we analysed the characteristics (accessibility, charge, polarity and secondary structure) of the environments of these lysine residues to assess how these might influence the observed outcomes. Notably, this reactivity of the different population of lysines appeared to exhibit relatively weak underlying biases in their characteristics and showed distributions similar to those of lysines across the entire proteome (Supplementary Figs. S14 vs S15); this importantly revealed that the reactions we observed are likely to be widespread and not dependent on specific unusually biased environments. Some partial trends were seen. Acetamides formed by pathway C were more likely to occur at lysines that are more buried (as judged by AlphaFold-enabled pPSE (prediction-aware part-sphere exposure)[36]) and with higher predicted $pK_{aH}$ values compared to *N*-acyl modifications (Fig. 4h, i). For **4**, *N*-succinamide-modified lysines and *N*-acyl-modified lysines both showed two regions of $pK_{aH}$ ~ 10.5 and ~11.4 but significant ($p < 0.01$) differences in their distribution, with *N*-succinamide-modified lysines derived from ring-opening pathway B being more frequent around lysines of lower predicted $pK_{aH}$ ~ 10.5 (Fig. 4j). For SATA **1** and SATP **2**, the levels of identification of *N*-succinamide modifications resulted in no measurably significant bias compared to other modifications (Fig. 4h, i). Additionally, there were no substantial biases observed in the distribution of electrostatic potential (ESP) or secondary structure (Fig. 4h–j and Supplementary Fig. S14b, 14c, 14d, 14e). Notably, we also compared the intensities of peptides with ring-opened *N*-succinamide modifications in the proteome labelled with **4** at 1 h and 24 h, revealing a significant decrease over time (whilst the total identified peptides showed only a slight decrease); this further confirmed the instability of ring-opened products of NHS esters (Supplementary Fig. S16).

Next, we sought to explore the effect of the acylation reagent leaving group to provide possible alternative strategies for mitigating the unwanted side-reactions that we have observed here. First, given that pathway B reactivity involves reaction of the succinimide ring we tested whether structural modification of the succinimide ring itself might affect reaction pathway outcome (Supplementary Fig. 17). We synthesised (Supplementary Figs. 5, 6) corresponding sulfonylated-NHS (NHSS) variants of **1** and **6**, SATA-NHSS **7** and Sia-Gal-Glc-NHSS **8**, respectively. Second, we also synthesised the corresponding pentafluorophenyl ester variant of **6** (Sia-Gal-Glc-S-PFP, **9**) within which the succinimide ring is wholly absent. Consistent with modulation of ring-opening reactivity, in comparison (Fig. 5a) to the product ratio observed with compound **6**, where pathway B *N*-succinamide derivative dominates and *N*-acyl derivatives are minor, the presence of a sulfo-group on the succinimide ring of **8** reduced unwanted reactivity leading to ~1:1 pathway A:B. Furthermore, consistent with this more limited ring-opening reactivity SATA-NHSS **7** yielded a similar mixture of pathway A and C products as had **1** (Supplementary Fig. 17). The PFP ester **9**, which does not possess a succinimide ring, afforded only *N*-acyl product (Fig. 5b) further confirming the identity of the products observed from **6** and **8** and importantly highlighted its strong utility as an alternative acylating reagent. Sulfo-analogues of NHS are commonly used for suggested increased solubility[12,37]. Together, our data here now suggests that their use may also confer the unanticipated added benefit of reducing unwanted pathway B ring-opening.

At the time of writing there were >2500 commercially-available NHS esters bearing the motif that we initially identified as driving unwanted pathway B ring-opening reactivity [2649 commercially available thio-NHS acetates (https://scifinder-n.cas.org/search/substance/63cd80bd33a83d49c9e5882a) and 1391 commercially available thio-NHS propanoates (https://scifinder-n.cas.org/search/substance/64189cfcc6a70a2a55a8b58b/1)] representing 7.3% of the total 55,410 commercially-available NHS esters (https://scifinder-n.cas.org/search/substance/65dc9e1903abfb1da11a351b/1), with this figure continuing to rise. Therefore, a complete screen of NHS esters proves prohibitive. When we tested some other common commercially-available NHS or NHSS esters (Supplementary Fig. 18) selected for their diverse functional use as either protein cross-linking reagents[38,39] or for the attachment of reactive groups (such as DBCO[40] or maleimide[41,42]) only pathway A acylation was observed. Therefore, whilst we have not tested all motifs that might drive pathway B ring-opening, these results for now confine our surprising observations to the thioalkanoyl $R-S(CH_2)_nCOO$ motif.

## Discussion

Whilst the use of NHS esters is long described for many decades, a survey of the literature suggests only few characterised observations of non-innocent NHS activity prior to our observations. Our results now surprisingly demonstrate that the ring-opening reaction of succinimide to generate various *N*-succinamide derivatives may be a commonly-occurring side reaction, even the prevailing product, for certain NHS esters. When combined with our data, these handful of scattered examples in the literature over more than four decades identify putative chemical driving forces of reduced acyl susceptibility to nucleophilic addition-elimination substitution that can prove sufficient to generate significant competing non-innocent NHS activity in amide formation from certain NHS esters: (a) hindered e.g. neopentyl-like[43], tertiary[44] or aryl[45] alpha carbons; and (b) potential additional electronic deactivation through π-conjugation[45]. These are further supported by observations, albeit under very different conditions, in peptide coupling that have attempted to exploit the NHS esters of L-Pro (or related tertiary *N*,*N*-dialkyl variants) in amide forming reactions with hindered nucleophiles[46–50] Nonetheless, the effect of α or β-thiosubstitution in the thioalkanoyl motif that we find here is highly surprising in that whilst mild retardation of acid-catalysed hydrolysis is known[51] there is not expected to be as large a perturbation [as measured by Taft (EtSCH−: $E_s = -0.47$, $\sigma^* = -0.56$ *cf* $CH(CH_3)_2−$ $E_s = -0.47$, $\sigma^* = -0.19$; $C(CH_3)_3−$ $E_s = -1.54$, $\sigma^* = -0.3$; Ph− $E_s = -2.55$, $\sigma^* = 0.6$) or Charton[52] parameters]. We therefore cannot discount a non-traditional mechanism, such as a direct participatory role of the thio substituent in activation of the imidic carbonyl[53].

Key features exist in protein chemistry that may complicate direct comparison with observations made in other systems and highlight the importance of using 'real-world' protein samples and associated conditions over e.g. small-molecule model systems. First, concentrations of substrate biomolecule are typically orders of magnitude lower and NHS-esters used in greater excess. Second, in proteins competition from non-Lys nucleophilic amino acid side chains exists on protein surfaces; NHS esters may therefore react with other residues to create labile linkages in aqueous solution, prone to hydrolysis or further acyl

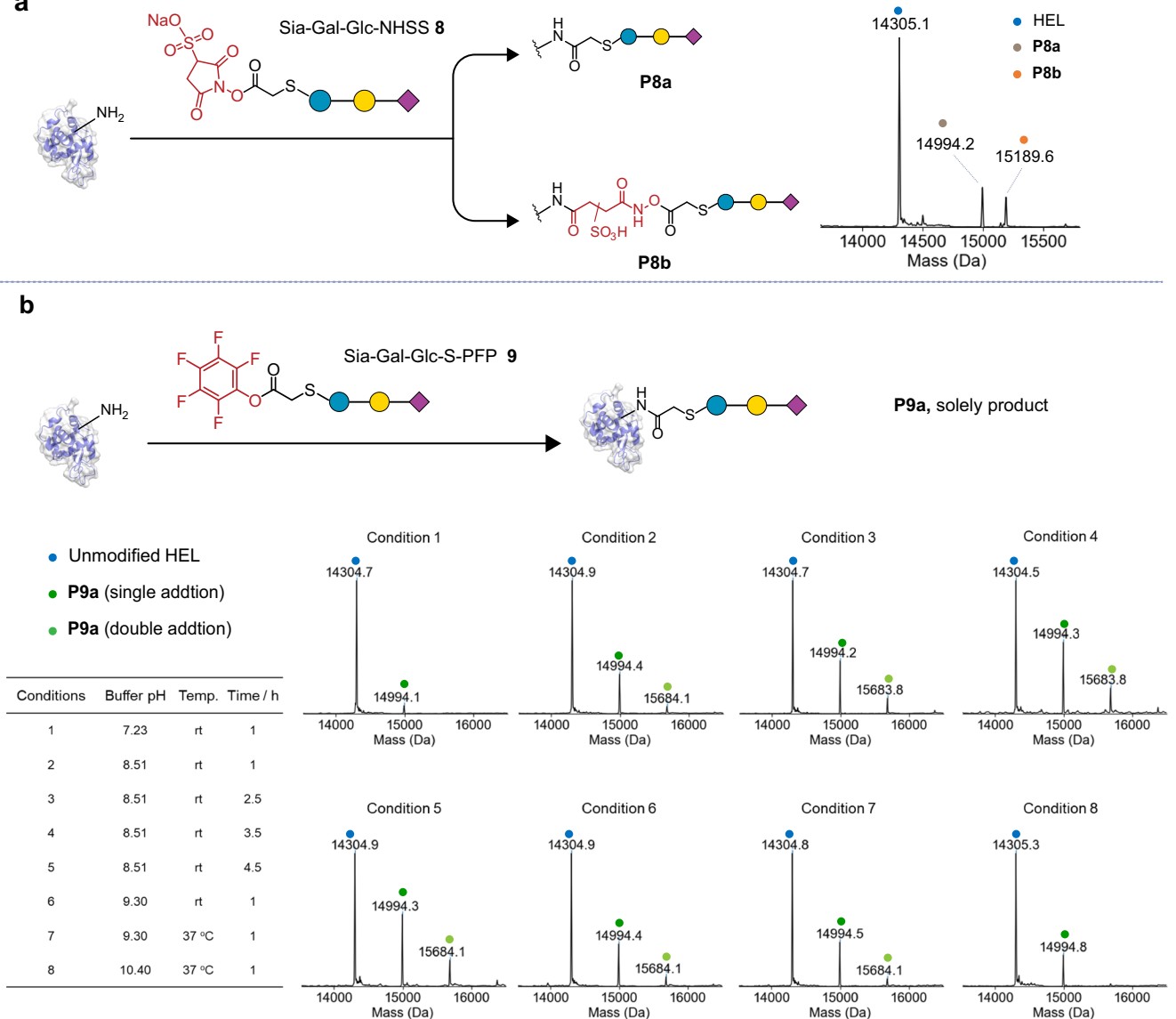

**Fig. 5 | Mitigation of ring-opening reactivity by use of NHSS and PFP reagents. a** LC-MS analysis of reactions of hen egg lysozyme (HEL) with Sia-Gal-Glc-NHSS **8**. **b** LC-MS analysis of reactions of HEL with Sia-Gal-Glc-S-PFP **9**. Reagents and conditions of (**a**) and (**b**): 200 equivalents of **8** or **9** (with respect to Lys), 1 mg/mL HEL, 0.1 M NaHCO₃, r.t, 2 h.

transfer[8,9,54,55]. Potentially, therefore, reactions with nucleophiles found in e.g. cysteine, tyrosine, histidine, serine and threonine side chains may divert normal reactivity[8,56].

Regardless of its origins, the thioalkanoyl motif that drives side-reaction has been used, for example, in the generation of platforms for antibody-drug conjugates as clinical candidates[27,57,58] and vaccines[59]. There are therefore several potentially important consequences of our observations:

i) the acylated hydroxamic esters that are formed by pathway B remain as activated esters[60] and so could give rise to slower but functional transacylation or hydrolysis in any subsequent uses of isolated bioconjugates containing such linkages. This may give rise to intramolecular transfer of payload, intermolecular covalent modification of bystanders or simply loss of conjugation (and so lowered 'loading density/copy number/drug:antibody ratios). Indeed, this is consistent with our observations here of instability that gives rise to a reduction of loadings over time observed by both intact protein MS and proteomic analysis (Fig. 2g, Supplementary Figs. 10, 11 and 16). The associated half-lives

for instability that we have observed here suggest that, whilst it is likely that the stability is different for different reagents, this mode of payload release has the potential to be a problem in many systems.

ii) the creation by pathway B of an extended succinamide linker generates a potentially altered or unwanted immunogenic site, increasing the likelihood of generating neutralising anti-drug antibodies (ADAs) and neoepitope hypersensitivity reactions[61,62]. The additional presence of pathway C for **1** and **2** may similarly have relevance in the context of pathologies associated with anti-acetylated-antibody responses[63];

iii) it also alters linker lengths, potentially confounding analyses in cross-linking-mediated structural measurements[64–66]; and

iv) as we observe here, low observed global levels of side-product may in fact mask locally-enhanced site-dependent changes in reactivity leading to side-product formation orders of magnitude higher and so evade detection by standard methods. An initial survey of the possible confounding effects of the use of resulting bioconjugates to probe biology suggests that they may be quite common [183

examples of 988 at time of writing (https://scifinder-n.cas.org/search/reference/65e799638359c55675bf807a)].

Lys modification chemistries still provide a primary approach for installation of protein modifications; it is a dominant industrial mode for bioconjugation[67]. Although our initial analyses of altered NHS-ester chemistries now reveals pathway B as viable, it will likely not have identified all of the effects driving such ring-opening; others from the many available NHS esters may well show the same such non-canonical reactivity. Our observation here also of the mitigation of this unwanted side-reaction through the use of alternative reagents suggests possible solutions to this problem. We therefore propose that the application of NHS esters in bioconjugation should be re-evaluated on a broader scale using the more precise and semi-agnostic characterisation methods that we have employed here.

## Methods

### Reagent and materials
Unless otherwise noted, the chemicals and solvents used were of analytical grade and were used as received from commercial sources. Lysozyme from hen egg white (lyophilised powder, ≥40,000 units/mg protein, Cat#L6876), DL-Dithiothreitol (Cat#D9779), SATA (≥95%, Cat#A9043), SATP (≥95%, Cat#10859), Fmoc N-hydroxysuccinimide ester (≥98%, Cat#46920), Succinimidyl 4-(N-maleimidomethyl)cyclohexane-1-carboxylate (SMCC, ≥97%, Cat#M5525), Sulfo-SMCC (Cat#M6035), Dibenzocyclooctyne-N-hydroxysuccinimidyl ester (DBCO-NHS ester, Cat#761524), Suberic acid bis(N-hydroxysuccinimide ester) (DSS, Cat#S1885), *Neisseria meningitides* CMP-sialic acid synthetase (Cat#C1999), *Pasteurella multocida* Sialyltransferase (Cat#S1951) were purchased from Sigma Aldrich.

### General considerations
All chemical reactions were carried out under an inert atmosphere using argon or nitrogen gas. All glassware was heat-dried unless aqueous chemistry was involved. Thin layer chromatography (TLC) was carried out using Merck silica-aluminium plates, with UV light (254 nm) and potassium permanganate, anisaldehyde or $H_2SO_4$ for visualisation. Column chromatography was performed using Merck Geduran® Si 60 silica gel or a Biotage Flash Purification System with a Kinesis Telos column. Room temperature refers to 20–25 °C. NMR data was obtained using Bruker AVIIIHD 400 MHz, Bruker AVII 500 MHz machines and Bruker Avance NEO 600 MHz machines. Reference values for residual solvents: $^{1}$H NMR- 7.26 (CDCl$_3$), 4.79 (D$_2$O) 3.31 (MeOD- D4) ppm, $^{13}$C NMR- O = 77.2 (CDCl3), 49.0 (MeOD- D4) ppm. Where appropriate, COSY, HSQC experiments were carried out to aid assignment. Coupling constants (J) are given in Hz and are uncorrected. NMR data was analysed using Mestrenova (V-11.0.1-17801). For full details of the synthesis of reagents and characterisation see the Supplementary Information file. Mass spectroscopy data was collected on Agilent 6120 Quadrupole spectrometer (ES), Waters LCT Premier (ES) instruments and a Xevo G2-S Q-ToF mass spectrometer coupled with a Waters Acquity UPLC system.

### Preparation of the protein conjugates with NHS esters
For lysozyme protein modification, a stock solution of lysozyme (5 mg/mL) was prepared by dissolving 25 mg lysozyme (1.7 µmol; 10.2 µmol NH2 groups) in 5.0 mL PBS (pH 7.2), PBS (0.1 M bicarbonate, pH 8.4) or 0.1 M bicarbonate solution (pH 8.5). NHS esters were prepared as a 10 mg/mL stock by dissolving into 0.2 mL dry DMSO. For the coupling reaction, specific equivalents of NHS esters were added to a specific concentration of HEL solution. The mixture was incubated at 0 °C, r.t., or 37 °C for a specific time. After reaction, the mixture was desalted to remove the excess of low molecular weight reagents. Subsequently, the NHS-modified lysozyme was determined by analysis of an aliquot of the crude mixture by LC-MS.

For Herceptin modification, before LC-MS analysis, a small aliquot (1 µL) of an antibody sample was diluted to 0.05 mg/mL by initially adding an aliquot of a stock DTT solution (80 mM) to give a final concentration of 8 mM and the rest made up with a 150 mM ammonium acetate pH 8.5 solution.

### Intact protein mass spectrometry and data analysis
Intact protein mass spectrometry was performed on Waters QTof mass spectrometers (Xevo G2-XS or Xevo G2-S) coupled to Acquity UHPLC systems. A Thermo Proswift column (250 mm × 4.6 mm × 5 µm) with a flow rate of 0.300 mL min$^{-1}$ and a solvent system of water +0.1% formic acid (solvent A) and acetonitrile +0.1% formic acid (solvent B) were employed for a total run time of 10 min with gradient elution as shown in Supplementary Table 2:

Nitrogen was used as the desolvation (650 L/h) and cone (30 L/h) gas with the following instrument parameters: capillary voltage 3 kV, cone voltage 20 V, source temperature 100 °C, desolvation temperature 400 °C and collision energy 6.0 eV.

The raw spectra of multiple charged ion series were deconvoluted using Waters MassLynx software (v4.1) and its maximum entropy (MaxEnt1) function with the following set-up: resolution 1.00 Da/channel, width at half height (protein-dependent i.e. 0.80 Da for HEL), minimum intensity ratios 33% left and right and an iteration to convergence. The deconvolution mass range is protein dependent i.e. 10,000–20,000 Da for Lysozyme. Reaction conversions were calculated as the relative intensity ratio of the peak of interest against the sum for all peaks.

For generating the mass spectra and data processing, the RAW files were deconvoluted using UniDec (Version 6.0.3)[68] and its UniChrom with the following set-up: m/z from 250 to 2500; Charge range: 1–50; Mass range: 10,000–20,000 Da; Sample Mass Every (Da): 0.1; Peak FWHM (Th): protein-dependent i.e. 0.80 Da for HEL.

### Preparation of LC-MS/MS sample of site-profiling
For digestion, ~5 µg of each sample in PBS with 2% SDS, 10 mM TCEP, 50 mM Chloroacetamide was precipitated onto 200 µg carboxylate coated paramagnetic beads (Sera-Mag Speed Beads, Cytiva) by addition of neat MeCN (80% final concentration). The beads were washed according to published SP3 protocol[69]. Beads were covered with 50 µL of 50 mM triethylammonium bicarbonate buffer with 0.2 µg of Lys-C (Wako) and Trypsin (Promega) and incubated at 37 °C for 4 h at 400 rpm. After collecting the supernatant with resulting peptides, beads were washed with 20 µL of 2% DMSO and combined with the supernatant. The mixture was acidified with neat formic acid to 5% final concentration.

### Quantification of peptide signal
Quantification of peptide signal with each of the modifications was calculated using the IonQuant module of the FragPipe suite with standard quantification settings; match between runs was enabled for searches of modifications using reagents 3–6[70]. Initial ratios of products of different pathways were calculated based on relative intensities of modified peptide ions within each sample for the most abundant of HEL peptides: CELAAAMK(mod)R. A similar method was used to assess the product ratio of different pathways on each of the lysine residues when using SATA and SATP reagents, 1 and 2. Only equivalent peptide ions (Same sequence and other modifications, if any) were considered for each of the lysine residues.

### Proteomic analysis of the coupling reaction on native E. coli proteome
*Native E. coli proteome extraction and reaction: E. coli* K12 strain was cultured in LB medium until reaching an optical density (OD$_{600}$) of 0.6–0.8. Cells were harvested by centrifugation at 4000 × *g* for 10 min at 4 °C, and the cell pellet was resuspended in lysis buffer (50 mM Tris-

HCl, pH 7.5, 150 mM NaCl, 1 mM EDTA, and a protease inhibitor cocktail). Cell lysis was performed using an ultrasonic cell disruptor, followed by protein concentration determination using a BCA protein assay kit. Reactions were conducted by incubating the lysate with compounds **1** and **2** for 30 min, and with compound **4** for 1 h or 12 h. Finally, the reaction mixtures were stored at −80 °C for further analysis.

*SP3 clean-up and Trypsin digestion*: Samples were digested using a modified single-pot solid phase assisted sample preparation method (SP3)[69] using magnetic carboxyl coated magnetic beads (Cytiva) at 4:1 w/w ratio. Proteins were precipitated onto the beads by addition of acetonitrile to 80% final concentration and incubation for 30 min with shaking at 1000 rpm. The beads with immobilised protein were then washed with 80% Ethanol three times, followed by three washes with 100% acetonitrile before incubation with digestion buffer (50 mM TEAB), containing sequencing grade Trypsin (Promega) at 1:25 enzyme-to-protein ratio for 4 h. The supernatant was collected, and beads were washed with 2% DMSO solution, which was collected and combined with the supernatant before acidification with formic acid to final concentration of 5% and centrifugation at $16,000 \times g$ for 10 min. The peptides were then desalted on Oasis HLB cartridges and eluted with 50% acetonitrile in ultrapure water.

## Chromatography and mass spectrometry

Peptides were separated on Ultimate 3000 RSLCnano system (Thermo Fisher Scientific) equipped with a C-18 PepMap100 trap column (300 µm ID × 5 mm L, 100 Å, Thermo Fisher Scientific) and an in-house packed Reprosil-Gold C-18 analytical column (50 µm ID × 500 mm L, 1.9 µm particle size, Dr. Maisch). Mobile phases (A: 0.1% FA, 5% DMSO and 94.9% water; B: 0.1% FA, 5% DMSO, 94.9% ACN) were delivered at a flow rate of 100 nL/min with a 15 min (10–38%B) or 120 min (8–28%B) gradient for single protein and lysate analysis, respectively. Eluting peptides were electro-sprayed into an Orbitrap Exploris 480/Orbitrap Fusion Lumos/Tribrid Ascend mass spectrometer (Thermo Fisher Scientific), using Data-Dependent Acquisition mode (DDA). Full MS scans (350–1400 m/z range) were acquired in the Orbitrap mass analyser at 60,000 resolution ($1.2 \times 10^6$ AGC target, 123 ms maximum injection time). Twenty or forty most intense precursors (charge states 2–7) from MS1 scans were selected and isolated at 1.2 Th with the quadrupole for fragmentation using higher-energy collision dissociation (HCD or CID). MS2 scans were acquired in the Orbitrap mass analyser at 7500 resolution ($2 \times 10^4$ AGC target, 32 or 64 ms maximum injection time).

## MS/MS data analysis

General setup of analysis software: Single protein datasets were searched in MSFragger[31,32] (v. 3.7) against a contaminants database containing lysozyme sequence (336 entries) with decoys generated in FragPipe interface (version: 19.1): MSFragger (version: 3.7), Philosopher (version: 4.8.0), IonQuant (version: 1.8.10); and Python. For the datasets produced from E. coli cells, the FASTA database was downloaded from the Uniprot (ID: UP000000625) on 2022-09-07, decoys and contaminant were added within FragPipe interface (version: 22.0): MSFragger (version: 4.1), Philosopher (version: 5.1.1), IonQuant (version: 1.10.27), Python and diaTracer (version: 1.1.5).

*Analysis of the mass of modifications with FragPipe*: Exploratory open searches were conducted using the Open Search functionality of MSFragger[31] and PTM Shepherd module[71]. The following settings were used: Precursor mass tolerance −150 to 500 Da, fragment mass tolerance 20 ppm, Calibration and Optimisation 'Mass calibration, parameter optimisation' enabled, Isotope Error '0', cleavage 'enzymatic', Clip N-term N enabled, enzyme name 'stricttrypsin', cut after 'KR', but not before 'P', missed cleavages '2', peptide length 7–50, peptide mass

range 500–5000 Da, methionine oxidation with max. 3 occurrences and acetyl N-terminal with max. 1 occurrence were set as variable modifications. Carbamidomethyl cysteine was set as a fixed modification, restrict delta mass to KSTY. All other options were left at the standard settings. Crystal-C was enabled. The FragPipe workflow with these parameters has been deposited in the PRIDE upload as 'fragpipe_open_final.workflow'.

For downstream data analysis, the 'global.profile.tsv' file was loaded and the values for the number of PSMs were plotted against the 'Mass Shift'.

Site assignment of modified peptides: The closed searches were conducted using default closed search parameters. The following settings were used: Precursor mass tolerance −20 to 20 ppm, fragment mass tolerance 20 ppm, 'Mass calibration, parameter optimisation' enabled, Isotope Error '0/1/2', cleavage 'enzymatic', Clip N-term N enabled, enzyme name 'stricttrypsin', cut after 'KR', but not before 'P', missed cleavages '2', peptide length 7–50, peptide mass range 500–5000 Da, methionine oxidation with max. 3 occurrences and acetyl N-terminal with max. 1 occurrence were set as variable modifications. Variable modifications on Lys were set to the masses indicated in Supplementary Table 1 with max. 2 occurrences. Carbamidomethyl cysteine was set as a fixed modification. All other options were left at the standard settings. Crystal-C was disabled. The FragPipe workflows with these parameters have been deposited in the PRIDE upload as 'fragpipe_closed_K_final.workflow'.

## pPSE prediction from AlphaFold2

The open-source Python package (StructureMap, https://github.com/MannLabs/structuremap) was used for integrating information (pPSE values, intrinsically-disordered region and secondary structure annotations) from predicted protein structures deposited in the AlphaFold database with proteomics data. The pPSEs for side-chain exposure estimation were calculated using a distance threshold of 12 Å and an angle of 70°. Amino acids with a pPSE ≤ 5 were considered to have a high exposure.

## pKa prediction from PDB Files

pKa values for lysine residues were predicted using the open-source PROPKA3 package[72] to assess local chemical environments in protein structures. The CIF files of predicted protein structures from the AlphaFold database were converted to PDB format using PDBFixer (https://github.com/openmm/pdbfixer). PROPKA3 was then run on the PDB files to calculate the pKa values of lysine residues. The resulting.pka files were processed using a custom Python script to extract lysine pKa values, residue number and chain ID.

## Electrostatic potential calculation using APBS

Electrostatic potentials for lysine residues were calculated using APBS[73]. PDB files were converted to PQR format with pdb2pqr using the AMBER force field and a pH of 7.0. Custom APBS input files were generated with the following parameters: Grid dimensions: 97 × 97 × 97 (for better resolution); Grid lengths: 150 Å in each direction; Dielectric constants: 2.0 for the solute (protein) and 78.0 for the solvent (water); Solvent radius (srad): 1.4 Å; Temperature: 298.15 K. The input files were configured to calculate the electrostatic potential using the linear Poisson-Boltzmann equation (LPBE) and the Smoothed Molecular Surface model (srfm). APBS was then run on the generated PQR files to produce.dx files containing electrostatic potential values.

## Reporting summary

Further information on research design is available in the Nature Portfolio Reporting Summary linked to this article.

## Data availability

The mass spectrometry data for proteomics and modification site identification generated in this study have been deposited in the ProteomeXchange Consortium through the PRIDE[74] partner repository, with the dataset identifier PXD056955 and PXD056949. Source data are provided with this paper. Any further data supporting the findings of this study are available from the corresponding author upon request. Source data are provided with this paper.

## Code availability

Dataset output from Fragpipe was further analysed with custom Python and R scripts, available on Zenodo at https://doi.org/10.5281/zenodo.13956916.

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

## Acknowledgements

Next Generation Chemistry at the Rosalind Franklin Institute has been supported by grants from UKRI-EPSRC (EP/V011 359/1, EP/T012 021/1, EP/X527 245/1).

## Author contributions

W.L., A.K., Y.D., S.M. and B.D. conceived the project and designed the experiments. W.L. and A.K. carried out the synthesis and protein conjugation. W.L. and Y.D. performed the proteomics experiments and mass spectrometry analyses. W.L. performed the bioinformatics analysis. All authors contributed to data interpretation. W.L. and B.D. wrote the manuscript with input from all authors.

## Competing interests

The authors declare no competing interests.
