## [Peer Review file · Nature Communications]

Thio-NHS esters are non-innocent protein acylating reagents

Corresponding Author: Professor Ben Davis

A version of this paper was originally rejected for publication by Nature Communications, however that decision was reconsidered after appeal by the authors.

Version 0:

Reviewer comments:

Reviewer #1

(Remarks to the Author)

The paper described by Davis and colleagues raised concerns about the reactivity of traditional amidation reactions with NHS esters. In their study, they found that alpha and beta-thio-NHS esters did not exhibit the typical acylation reactivity towards lysine in proteins. Instead, they observed several side reactions, including ring-opening of succinimide and N-acylation with thioester, which cast doubt on the general applicability of NHS esters for lysine bioconjugation. However, other NHS esters tested in their study (Supplementary, Fig 14) without alpha-sulfur or beta-sulfur activation did not show any side reactivity other than acylation. Therefore, this unusual reactivity for thio-NHS esters seems not to be a significant concern for other NHS ester chemistries, and it would be important only for researchers working with those specific esters. I would not recommend the publication of this paper in Nature Communications unless the effect can be better understood, and ideally other types of activated NHS esters can be identified having a similar issue. In particular, other alpha/beta heteroatom effects could be investigated. Ideally, the authors would gather a more complete understanding and be able to predict at least which structures should be a matter of concern. In addition, this could be completed by the suggestion of the "best alternative reagent" for further use in bioconjugation.

Suggested further revisions.

1. Detailed studies on small molecules (lysine or lysine-containing peptides with thio-NHS esters) should be conducted, as they will provide insights to understand the original reactivity between thio-NHS esters and free amines without a protein environment, and see how much the latter is important for side reactivity. This may help also to better understand the fundamental origin of the observed reactivity and investigate further derivatives (other heteroatoms in alpha or beta position, thio in gamma position). Also the data in Figure 1b seems to suggest that electronics on the sulphur atom is important, with more side reactivity observed with thioesters than thioethers. To complete this trend, it would be interesting to have also a thioether in beta position in the figure. It would be also interesting to further understand the increase of side reactivity with more complex substituent (steric effect? is that only specific to proteins or general?) It would be also interesting to add some thioether to the reaction mixture to see if an intermolecular effect is possible.
2. More stability test with the isolated ring-opened product will be of great interest. The authors just mention some qualitative studies on the stability of the mixture of adducts obtained. Such stability studies will definitively establish if the adducts could be useful or not.
3. The title should be modified, as only thio-NHS esters were shown to undergo the ring-opening reactions, other NHS ester derivatives showed good acylation selectivity.
4. There is only one figure in the main text, but 15 figures in the supporting information. For reading convenience, the authors should include more figures in the main text instead of putting them to the supporting information, especially for important information like Supplementary Figure 10. In fact, Figure 10 proposes a partial solution of the side reactivity issue. Ideally, the authors could consolidate their study and propose the best NHS-derivative to use to avoid side reactivity in the main part of the manuscript. The work would be significantly strengthened by such a validated alternative for bioconjugation.

Reviewer #2

(Remarks to the Author)

I co-reviewed this manuscript with one of the reviewers who provided the listed reports. This is part of the Nature Communications initiative to facilitate training in peer review and to provide appropriate recognition for Early Career

Researchers who co-review manuscripts.

Reviewer #3

(Remarks to the Author)

The authors investigated the inherent properties of widely used N-hydroxysuccinimide (NHS)-ester-based bioconjugation reagents. The three electrophilic centers in the NHS-activated thioalkanoyl substructure of SATA and SATP could lead to three distinct pathways in parallel. It includes (A) NHS-ester-mediated acylation, (B) succinimide ring opening, and (C) thioester-mediate acylation.

The study is extended from SATA and SATP to S-benzyl-NHS-ester, Glc-NHS, Gal-Glc-NHS, and Sia-Gal-Glc-NHS. The extent of contributions from each pathway is further investigated by MS. The results indicate that the pathway, chemoselectivity, and regio-selectivity indicate dependency on the sites.

While the observations on chemoselectivity attributes are interesting and present a note of caution for the reagent design, they do not offer any solutions. The latter would be exploratory, and it is difficult to assess if it could present results interesting enough to warrant publication in Nature Communications. Hence, the manuscript is suitable for a more specialized journal.

Additionally, the authors should consider a discussion on the following points before resubmitting to another journal:

- (a) A substantial amount of data is in the supplementary information. The authors could consider redrafting the manuscript.
- (b) The reaction with HEL is performed with excess bioconjugation reagent. Since NHS ester is highly reactive, what is the basis for such stoichiometries? In other words, is the reaction biased towards product distribution between different reaction pathways?
- (c) While the reagent derivatives offer differences in chemoselectivity, their design does not offer meaningful control over a specific pathway. How can that be resolved?
- (d) Although the method has been examined with Herceptin, verification with structurally diverse proteins could add value.

Reviewer #4

(Remarks to the Author)

Version 1:

Reviewer comments:

Reviewer #1

(Remarks to the Author)

This revised manuscript provides a detailed examination of the reactivity between thio-NHS esters and lysine residues in various proteins, extending the study to proteome-wide applications. The improved organization has significantly enhanced the paper's readability and logical flow. Consequently, this new version is greatly enhanced and I recommend publication in Nature Communications after addressing the following minor points:

1. I recommend removing the structures of compounds 3–6 from Figure 1, as they are not directly relevant to the data presented in that figure. Also, these structures are already included in Figure 2. Based on all the data the author presented. I have reason to believe that the undesired ring-opening and acetamide formation product arises from the intrinsic reactivity between S-NHS ester and free amines. To confirm this, I still recommend that the authors investigate this reactivity at the small molecule level. If this intrinsic reactivity is verified, it would help alleviate concerns about similar issues occurring with other types of NHS ester.
3. For the title, I would still suggest changing to a more specific NHS ester like "Thio-NHS ester are non-innocent protein acylating reagents", as there are no evidence showing that other types of NHS ester share similar reactivity,

Reviewer #2

(Remarks to the Author)

Reviewer #4

(Remarks to the Author)

The authors identified the side-products resulting from NHS labeling across a diverse range of proteins, including the 'ring-opening' of the succinimide moiety within the NHS ester. While the findings are interesting, further data are required to enhance the impact of this study. For instance, additional biological validation of the binding sites, as well as the application of this approach in live cells and cell lysates, which could expand the scope of druggable proteome.

Comments to reviews

Reviewer #1 (Remarks to the Author):

The paper described by Davis and colleagues raised concerns about the reactivity of traditional amidation reactions with NHS esters. In their study, they found that alpha and beta-thio-NHS esters did not exhibit the typical acylation reactivity towards lysine in proteins. Instead, they observed several side reactions, including ring-opening of succinimide and N-acylation with thioester, which cast doubt on the general applicability of NHS esters for lysine bioconjugation. However, other NHS esters tested in their study (Supplementary, Fig 14) without alpha-sulfur or beta-sulfur activation did not show any side reactivity other than acylation. Therefore, this unusual reactivity for thio-NHS esters seems not to be a significant concern for other NHS ester chemistries, and it would be important only for researchers working with those specific esters. I would not recommend the publication of this paper in Nature Communications unless the effect can be better understood, and ideally other types of activated NHS esters can be identified having a similar issue. In particular, other alpha/beta heteroatom effects could be investigated. Ideally, the authors would gather a more complete understanding and be able to predict at least which structures should be a matter of concern. In addition, this could be completed by the suggestion of the "best alternative reagent" for further use in bioconjugation.

- We note the reviewer's suggestion that we investigate other heteroatom systems and have sought to understand this further.
- The valency of the heteroatom is, of course, vital. What sets the sulfur heteroatom apart is that, as a thioether, it can act as a strong Lewis base and, therefore, a nucleophile (and nucleophilic catalyst) in a manner that is not typically irreversible. We would welcome suggestions on what specifically these further studies might entail, if they could be set in the context of a mechanistic hypothesis.
- The referee asserts that our results would not be a significant concern for NHS esters chemistry of other types.
- We have revised our Discussion section, where we highlight that these type of NHS ester reagents are already major players in the global context of such chemistry (they are not minor species) but in fact used widely across both academia and industry. Therefore, the impact, even in this case set is important and widespread.
- Notably the number of database NHS esters that have the putative motif to drive unwanted pathway B ring-opening reactivity has markedly increased even over the past months; we have updated the latest data here.
- SATA and SATP are some of the most widely used agents for protein modification from the NHS family (see our revised analysis of this in the Discussion).
- In addition, our mechanistic investigations now into side-reaction mechanisms reveal that steric hindrance and electronic effects influencing acyl group reactivity, as well as lysine reactivities dependent on the protein micro-environment, may play roles in driving ring-opening reactions. This suggests that other types of NHS esters, not just those in our study, could undergo similar side reactions, highlighting the need to avoid NHS esters to prevent unintended linkages.
- We have added further text to the revised manuscript emphasizing this point.
- We appreciate the referee's suggestion of highlighting another "best alternative reagent".
- In the revised manuscript, we now include in the main text and **Fig 5**, acylation chemistries that allow the same linkages to be achieved by using alternatives. These include, in particular, the use

of sulfonylation to moderate ring-opening and the utility of pentafluoropyridyl esters instead of NHS esters.

- We have added further text to the revised manuscript emphasizing this point.

Suggested further revisions.

1. Detailed studies on small molecules (lysine or lysine-containing peptides with thio-NHS esters) should be conducted, as they will provide insights to understand the original reactivity between thio-NHS esters and free amines without a protein environment, and see how much the latter is important for side reactivity. This may help also to better understand the fundamental origin of the observed reactivity and investigated further derivatives (other heteroatoms in alpha or beta position, thio in gamma position). Also, the data in Figure 1b seems to suggest that electronics on the sulphur atom is important, with more side reactivity observed with thioesters than thioethers. To complete this trend, it would be interesting to have also a thioether in beta position in the figure. It would be also interesting to further understand the increase of side reactivity with more complex substituent (steric effect? is that only specific to proteins or general?) It would be also interesting to add some thioether to the reaction mixture to see if an intermolecular effect is possible.

- We thank the referee for the suggestions of small molecule model systems and the use of intermolecular thioethers.
- The primary purpose of our study was to investigate the use of NHS esters under conditions directly relevant to bioconjugation. Small molecule models, by definition, therefore provide a less relevant (and sometimes misleading) test set.
- It should be noted that the concentration regimes under which bioconjugations are conducted typically exceed the dynamic range of concentrations for small molecule study and detection, limiting the insight these model systems can provide.
- We believe strongly that the best systems to work on are the very proteins that are the targets themselves, such as therapeutically-relevant antibodies that are used to create conjugates like Herceptin (see our manuscript) and others now added (see **Fig. 3** and below).
- Therefore, we are unsure what specific additional information would be gathered from working with lysine-based systems and what insights into reactivity might be gained that cannot already be obtained from protein systems and the advanced and comprehensive analytical methods that we have already employed here.

- The referee makes some comments that are technically inaccurate regarding the role of thioesters or thioethers. In fact, there is no direct trend between these in the study that we performed. The differences observed simply arise from different pathways, including an additional pathway available to thioesters but not to thioethers.
- Therefore, the ring-opening chemistry itself is balanced with others, and in this context, it is more likely that the thioether is important. We would not wish to oversimplify something that we believe is a complex reaction manifold.

- We also respectfully highlight to the referee that the inherent nature of the chemistries we study means that intermolecular reactions would require a significant increase in concentration to compete with intramolecular reactions.
- It is these intramolecular effects that are at the heart of what we are describing. By definition, in our mixtures we are already testing intermolecular competition by using an excess of the reagent.
- Therefore, the suggestion of intermolecular experiment may misunderstand that we are experimentally exploring a delicate balance between competing reactions with similar kinetics that

can lead to a diverse range of reactive outcomes. This is the essence of our study: the NHS ester reagent – something that is perceived to have a certain, defined reactivity – is, in fact, on a tipping point away from the ‘normal’ mode of reactivity i.e. it is non-innocent.

- We have added further text to the revised manuscript in the Discussion emphasizing this point.

2. More stability test with the isolated ring-opened product will be of great interest. The authors just mention some qualitative studies on the stability of the mixture of adducts obtained. Such stability studies will definitively establish if the adducts could be useful or not.

- This is an excellent suggestion and we have now explored this in detail.
- Specifically, we have observed that, after initial formation, pathway B ring-opening products (**P4-6b**) decrease with prolonged incubation (new **Supplementary Fig. 10, and 11**), with most **P4b**, for example, converting to a +115 Da fragment and exhibiting a half-life of ~3.4 days (**Fig. 2g**), indicating the instability of the extended succinamide linker – this appears also to decrease with increasing structural complexity.
- As the referee will appreciate, the crucial point here is that these conjugates are widely used in the generation of vaccines and biologics, sometimes even involving the attachment of cytotoxins. Any form of instability is likely to give rise to highly confounding effects in their use – this will be a major concern for the pharmaceutical industry for example through the non-desirable release of toxins.
- These results added in revision now suggest that these are likely issues that will be encountered from such NHS chemistry, further highlighting the impact of our study.
- The associated half-lives for instability that we have observed here suggest that, whilst it is likely that the stability is different for different reagents, this mode of payload release has the potential to be a problem in many systems.
- Notably, we even observed these instabilities within complex protein mixtures derived from entire proteomes. Thus, we compared the intensities of peptides with ring-opened *N*-succinamide modifications in a proteome labelled with **4** at 1 hour and 24 hours, revealing a significant decrease over time (whilst the total identified peptides showed only a slight decrease); this further confirmed the instability of ring-opened products of NHS esters (**Supplementary Figure S16**).
- We have added further text to the revised manuscript emphasizing these points.

- It should be noted that another concern that our data will importantly raise is the variation in the nature of the link itself. This specifically raises concerns regarding quality control and the identification of pure products i.e. as we now show, the desired products are not being produced in pure form by current methods and yet have not been declared and / or detected.
- These altered linkers also create potential epitopes that could trigger immunogenic responses, as we now discuss at greater length in our Discussion section. In fact, it is conceivable that some of these neoepitopes are already observed in certain clinical uses of such conjugates.
- Our raising of the awareness of this problem we think therefore will have broad and important consequences.
- We have added further text to the revised manuscript emphasizing this point.

- Finally, such side reactions of NHS esters will also lead to uncontrolled drug-antibody ratios (the DAR) in ADCs. The current assumption of “innocent” (reliable) reactivity used to control DAR must now be revisited based on our work, particularly when using SATA. Our data suggests that the DAR will be influenced by the side reactivity of SATA that we have discovered with further downstream consequences of uncontrolled toxicity.
- We have added further text to the revised manuscript emphasizing this point.

3. The title should be modified, as only thio-NHS esters were shown to undergo the ring-opening reactions, other NHS ester derivatives showed good acylation selectivity.

- We are happy to consider adjusting the nature of the title, but this was carefully chosen. We do not claim that all NHS are undergoing ring-opening reaction, but we simply seek to highlight that important ones do behave surprisingly i.e. they are not perfect reagents = ‘non-innocent’.

4. There is only one figure in the main text, but 15 figures in the supporting information. For reading convenience, the authors should include more figures in the main text instead of putting them to the supporting information, especially for important information like Supplementary Figure 10. In fact, Figure 10 proposes a partial solution of the side reactivity issue.

- As suggested, we have now moved critical and additional figures into the main text to change the balance of the paper.
- Specifically, we have combined the previous **Supplementary Figs. 7 and 9** from the SI into the current **Fig. 2a-e** in the main text and moved some abundance comparisons from original **Fig. 1b** to the current **Fig. 2f**. The current **Figs. 3 and 4** contain newly determined data, that also incorporates Herceptin data that came from the previous **Supplementary Fig. 15**.
- We thank the referee for their suggestion to improve the clarity of our work.

Ideally, the authors could consolidate their study and propose the best NHS-derivative to use to avoid side reactivity in the main part of the manuscript. The work would be significantly strengthened by such a validate alternative for bioconjugation.

- As suggested and as noted above, we have now included sections and figures (see, in particular, **Fig. 5**) that suggest possible alternative methods for acylation chemistry that could be applied.
- Specifically, we sought to explore the effect of the acylation reagent leaving group to provide possible alternative strategies for mitigating the unwanted side-reactions that we have observed here.
- First, given that pathway B reactivity involves reaction of the succinimide ring we tested whether structural modification of the succinimide ring itself might affect reaction pathway outcome (**Supplementary Fig. 17**). We synthesized (**Supplementary Fig. 5,6**) corresponding sulfonylated-NHS (NHSS) variants of **1** and **6**, SATA-NHSS **7** and Sia-Gal-Glc-NHSS **8**, respectively. Second, we also synthesized the corresponding pentafluorophenyl ester variant of **6** (Sia-Gal-Glc-S-PFP, **9**) within which the succinimide ring is wholly absent.
- Consistent with modulation of ring-opening reactivity, in comparison (**Fig. 5a**) to the product ratio observed with compound **6**, where pathway B *N*-succinamide derivative dominates and *N*-acyl derivatives are minor, the presence of a sulfo-group on the succinimide ring of **8** reduced unwanted reactivity leading to ~ 1:1 pathway A:B. Furthermore, consistent with this more limited ring-opening reactivity SATA-NHSS **7** yielded a similar mixture of pathway A and C products as had **1** (**Supplementary Fig. 17**).
- The PFP ester **9**, which does not possess a succinimide ring, afforded only *N*-acyl product (**Fig. 5b**) further confirming the identity of the products observed from **6** and **8** and importantly highlighted its strong utility as an alternative acylating reagent.
- Together, our data here now suggests that the use of both of these reagents may now provide alternatives to reduced or avoided unwanted pathway B ring-opening.

Reviewer #3 (Remarks to the Author):

The authors investigated the inherent properties of widely used N-hydroxysuccinimide (NHS)-ester-based bioconjugation reagents. The three electrophilic centers in the NHS-activated thioalkanoyl substructure of SATA and SATP could lead to three distinct pathways in parallel. It includes (A) NHS-ester-mediated acylation, (B) succinimide ring opening, and (C) thioester-mediated acylation.

The study is extended from SATA and SATP to S-benzyl-NHS-ester, Glc-NHS, Gal-Glc-NHS, and Sia-Gal-Glc-NHS. The extent of contributions from each pathway is further investigated by MS. The results indicate that the pathway, chemoselectivity, and regio-selectivity indicate dependency on the sites.

While the observations on chemoselectivity attributes are interesting and present a note of caution for the reagent design, they do not offer any solutions.

- Thank you, as we noted for referee 1, we have now included figures (in particular, **Fig. 5**) as well as a section in the main text that analyses possible solutions (see also above).
- One solution is to tune based on our mechanistic observations.
- The other solution is the total avoidance of NHS esters as reagents and the use of alternatives for forming amides such as pentafluorophenyl esters – we have demonstrated here that these are effective alternatives.
- We have added further text to the revised manuscript emphasizing these points.

The latter would be exploratory, and it is difficult to assess if it could present results interesting enough to warrant publication in Nature Communications. Hence, the manuscript is suitable for a more specialized journal.

- Our broad, cross-disciplinary study – the non-innocent nature of NHS ester chemistry (one of the most widely applied chemical reactions globally) will have significant implications across academia and the pharmaceutical industry.

Additionally, the authors should consider a discussion on the following points before resubmitting to another journal:

(a) A substantial amount of data is in the supplementary information. The authors could consider redrafting the manuscript.

- As suggested, and as noted for referee 1, we have now moved critical and additional figures into the main text to change the balance of the paper.
- Specifically, we have combined the previous **Supplementary Figs. 7 and 9** from the SI into the current **Fig. 2a-e** in the main text and moved some abundance comparisons from original **Fig. 1b** to the current **Fig. 2f**. The current **Figs. 3 and 4** contain newly determined data, that also incorporates Herceptin data that came from the previous **Supplementary Fig. 15**.
- We thank the referee for their suggestion to improve the clarity of our work.

(b) The reaction with HEL is performed with excess bioconjugation reagent. Since NHS ester is highly reactive, what is the basis for such stoichiometries? In other words, is the reaction biased towards product distribution between different reaction pathways?

- The referee raises an interesting question about the stoichiometry used. We have purposefully used conditions that are typical in the field in terms of an excess of reagent. This approach is customary in studies of bioconjugation and protein modification both academically and industrially and so importantly reflects on the non-innocent application of these reagents.
- We refer the referee to the general reviews that we cite in our introduction and are happy to provide further citations of this common practice and the standard (and so very relevant) nature of the conditions that we have employed.
- We have added further text to the revised manuscript emphasizing these points.
- It should be noted too that such excess makes no essential difference to the product distribution of the different reaction pathways. Such excess means that these reactions are essentially pseudo-first order in nature (the protein is by far the limiting reagent).
- In that context, there is no aspect of this work that is biased by the ratio of excess as the referee suggests.
- Analysis of the rate equations from such pseudo-first order reactions shows that chemoselectivity is not a measure that is inherently determined by reagent concentration but in fact one that is instead determined by the apparent rate constant associated with each term of the rate equation for a given reaction (i.e. each reaction branch).
- Thank you for raising this question.

(c) While the reagent derivatives offer differences in chemoselectivity, their design does not offer meaningful control over a specific pathway. How can that be resolved?

- Again, this reiterates a previously posed question (see also above).
- As we set out above, the manuscript proposes several methods of control. Either by tuning (see below) or via a very clear method for controlling the problem of side reactions – i.e. use of alternative acylation technologies.
- We now suggest, for the first time, that ring opening of the NHS moiety itself may be a prevalent reaction for some of the most important reagents that are used in the bioconjugation industry.
- The reason for varying the nature of the reagents was to explore first aspects of the mechanism: the position of the heteroatom (alpha versus beta), the nature of the sulfur moiety (ester versus ether), and the bulk of the attached group. By examining these factors, we aimed to understand their various influences on reactivity.
- These studies have provided us with insights into the mechanism, indicating that a combination of both bulk and electronic effects play a critical role in driving the unwanted ring-opening reaction.
- These observations now, in turn, now allow tuning. We have shown that here by the use of sulfonylated reagents to moderate ring opening pathway B and also the suggestion of alternative acylation chemistries that avoid it entirely.
- We have added further text to the revised manuscript emphasizing these points.

(d) Although the method has been examined with Herceptin, verification with structurally diverse proteins could add value.

- Exactly as suggested we have now greatly expanded the range of proteins within which these reactions might be being observed.
- Specifically, we have gauged reactivity in proteins containing more lysines (from 12 to 35), using a variety of representative protein folds and functions: the small α -helical nuclear histone H3

proteins from *H. sapiens* and *X. laevis*, the β -strand *C. jejuni* glycoprotein AcrA, the β -barrel green fluorescent protein GFP, and $(\alpha\beta)_8$ -barrel glycosidase enzyme SS β G. All showed ring-opening reactivity giving rise to pathway B *N*-succinamide derivative modification (**Fig. 3a** and **Supplementary Fig. 12**). Reactions (**Fig. 3b** and **Supplementary Fig. 13**) of **1**, **4** and **6** with Herceptin Fab (as a model therapeutic IgG antibody also mirrored our findings with other proteins: with **4** and **6** the unwanted ring-opening pathway B adduct was the major product. These results suggest that the occurrence of non-canonical pathways B and C (ring-opening or acetylation) are a more general phenomenon and not specific to a particular protein substrate.

- In addition, through the use of cellular proteomes we have been able to survey the occurrence of this mode of reactivity in > 1000 proteins, showing clear breadth of occurrence.
- Specifically, to more comprehensively understand whether the lysine micro-environment plays a critical role in driving such ring-opening reactions, we then profiled reactive lysines within extremely diverse complex protein samples. Thus, the proteomes of disrupted prokaryotic cells (released from their native contexts) were used to assess the environmental and dynamic reactivity of **1**, **2**, and **4**. *E. coli* (K12 strain) cells were lysed under physiological conditions and treated with **1**, **2**, and **4**, along with a control group treated with vehicle, followed by analysis for site modifications (**Fig. 4a**). Using an open search method, we identified the distribution of deviations from ring-opening pathway B modifications from **1**, **2**, and **4**, as well as pathway *N*-acyl modifications from **1**, **2**, and **4**, and acetamide pathway C modifications from **1** and **2**; these represented the major modifications occurring proteome-wide (**Fig. 4b, 4c, and 4d**). We then conducted a closed search to explore masses with the Δm_{exp} as the offset mass (see **Methods**). Following thorough data filtering, significant levels of ring-opening pathway B *N*-succinamide formation was observed (**Fig. 4e, 4f, 4g, and Supplementary Figure S14a**) in a background of pathways A and C. Notably, we also identified that a substantial number of lysine residues reacted through two or even three of these pathways (**Fig. 4e, 4f, 4g**).
- Notably, we also compared the intensities of peptides with ring-opened *N*-succinamide modifications in the proteome labelled with **4** at 1 hour and 24 hours, revealing a significant decrease over time (whilst the total identified peptides showed only a slight decrease); this further confirmed the instability of ring-opened products of NHS esters (**Supplementary Figure S16**).
- Note too, in key cases, we have not only already compared different proteins with different structures, as suggested, but also compared the reactivity of individual sites within those proteins.
- Our additional experiments therefore throw light not only onto protein diversity but also further address the mechanistic questions raised by the referee.
- This scoping of many different types of lysine environments has now provided an unusually detailed and revealing analysis. It highlights here how very different these many environments can be in their reactions with NHS ester reagents – this now importantly suggests that each environment should be treated individually.
- Specifically, the reactivity of the different population of lysines appeared to exhibit relatively weak underlying biases in their characteristics, including accessibility, charge, polarity, and secondary structure (**Supplementary Figure S14 vs S15**). However, some partial trends could be seen, lysines that undergo pathway C prefer to be buried and have higher pK_{aH} values compared to those that undergo pathway A (**Fig. 4h and 4i**). *N*-succinamide-modified lysines and *N*-acyl modified lysines showed similar distribution regions but significant differences in that distribution (**Fig. 4j**), suggesting greater ring-opening for lower pK_{aH} residues. The different reactivity of each lysine is therefore a result of a combination of factors – each environment should be treated essentially individually.
- We have added further text to the revised manuscript emphasizing these points.

REVIEWERS' COMMENTS

Reviewer #1 (Remarks to the Author):

This revised manuscript provides a detailed examination of the reactivity between thio-NHS esters and lysine residues in various proteins, extending the study to proteome-wide applications. The improved organization has significantly enhanced the paper's readability and logical flow. Consequently, this new version is greatly enhanced and I recommend publication in Nature Communications after addressing the following minor points:

- Thank you.

1. I recommend removing the structures of compounds 3–6 from Figure 1, as they are not directly relevant to the data presented in that figure. Also, these structures are already included in Figure

- Thank you for this advice – we would like to retain these as we believe that they aid clarity.

2. Based on all the data the author presented. I have reason to believe that the undesired ring-opening and acetamide formation product arises from the intrinsic reactivity between S-NHS ester and free amines. To confirm this, I still recommend that the authors investigate this reactivity at the small molecule level. If this intrinsic reactivity is verified, it would help alleviate concerns about similar issues occurring with other types of NHS ester.

- As discussed, the concentrations and kinetics at the small molecule level are very different to those at a protein level.
- We believe that our determination of reactivity in context is directly relevant.

3. For the title, I would still suggest changing to a more specific NHS ester like “Thio-NHS ester are non-innocent protein acylating reagents”, as there are no evidence showing that other types of NHS ester share similar reactivity,

- As suggested, we have revised this.

Reviewer #2 (Remarks to the Author):

- Thank you.

Reviewer #4 (Remarks to the Author):

The authors identified the side-products resulting from NHS labeling across a diverse range of proteins, including the 'ring-opening' of the succinimide moiety within the NHS ester. While the findings are interesting, further data are required to enhance the impact of this study. For instance, additional biological validation of the binding sites, as well as the application of this approach in live cells and cell lysates, which could expand the scope of druggable proteome.

- Thank you for this support and this suggestion – as we already describe in our revised manuscript this work has indeed already been validated in cell lysates, exactly as suggested.
- Please see the sections discussed in detail in page 10 and all of the associated data found in Figure 4 (as well as the Methods section).